Growth rate and locomotor performance tradeoff is not universal in birds

http://orcid.org/0000-0003-3922-5218 Zhao Tao 1 taozhao@nigpas.ac.cn
Li Zhiheng 2 lizhiheng@ivpp.ac.cn
1 State Key Laboratory of Palaeobiology and Stratigraphy, Nanjing Institute of Geology and Palaeontology and Center for Excellence in Life and Paleoenvironment, Chinese Academy of Sciences , Nanjing , China
2 Key Laboratory of Vertebrate Evolution and Human Origins of Chinese Academy of Sciences, Institute of Vertebrate Paleontology and Paleoanthropology and Center for Excellence in Life and Paleoenvironment, Chinese Academy of Sciences , Beijing , China
Garant Dany
Electronic publication date: 2020 Jan 23
Publication date: 2020
Volume: 8
Electronic Location ID: e8423
Received 2019 Sep 4; Accepted 2019 Dec 17
Copyright: © 2020 Zhao and Li
Copyright year: 2020
Copyright holder: Zhao and Li
License: This is an open access article distributed under the terms of the Creative Commons Attribution License, which permits unrestricted use, distribution, reproduction and adaptation in any medium and for any purpose provided that it is properly attributed. For attribution, the original author(s), title, publication source (PeerJ) and either DOI or URL of the article must be cited.
License URL: https://creativecommons.org/licenses/by/4.0/

Keywords: Birds, Growth rate, Flight ability, Flight muscles, Wing span, Wing area, Wing aspect ratio

Funding: National Natural Science Foundation of China 41688103 Chinese Academy of Sciences KC 217113 National Natural Science Foundation of China 41772013 The research was supported by the National Natural Science Foundation of China (41688103). Zhiheng Li was also funded by the Hundred Talents Program of the Chinese Academy of Sciences (KC 217113) and the National Natural Science Foundation of China (41772013). The funders had no role in study design, data collection and analysis, decision to publish, or preparation of the manuscript.

==============================
Though a tradeoff between growth rate and locomotor performance has been proposed, empirical data on this relationship are still limited. Here we statistically analyze the associations of growth rate and flight ability in birds by assessing how growth rate is correlated with three wing parameters of birds: flight muscle ratio, wing aspect ratio, and wing loading. We find that fast-growing birds tended to have higher flight muscle ratios and higher wing loadings than slow-growing birds, which suggests that fast-growing birds may have better takeoff performance, but lower efficiency in maneuvering flight. Accordingly, our findings suggest that the relationship between growth rate and flight ability is more complex than a simple tradeoff. Since the hindlimbs also contribute greatly to the locomotion of birds, future investigations on the relationship between growth rate and hindlimb performance will provide more insights into the evolution of birds.

Introduction

Growth rates vary considerably among different taxa and organisms (Arendt, 1997; Case, 1978; Ricklefs, 1973, 1968). It has been suggested that the growth rate for an organism results from a compromise between benefits and costs of rapid growth within physiological constraints (Arendt, 1997; Dmitriew, 2011). Rapid growth allows organisms to shorten the duration of reaching maturity when they are vulnerable to predators (Case, 1978). However, rapid growth may reduce longevity (Gabriela, 2018; Metcalfe & Monaghan, 2003) and reduce investment in other functions, because overall resources are limited (Arendt, 1997; Dmitriew, 2011; Martin et al., 2011). One of the main functions that are suggested to be negatively affected by rapid growth is locomotor performance (Billerbeck, Lankford & Conover, 2001; Dmitriew, 2011; Lee, Monaghan & Metcalfe, 2010). For example, Billerbeck, Lankford & Conover (2001) showed that within the Atlantic silversides (Menidia menida), the fast-growing fish have lower maximum prolonged and burst swimming speeds than slow-growing ones.

Birds are ideal for testing factors that are suggested to influence growth rate, because previous studies have accumulated relatively abundant data (Martin, 2004; Remeš & Martin, 2002; Starck & Ricklefs, 1998). By far, the variation of growth rate in birds has been found to be associated with a suite of factors. Growth rate is inversely correlated with body mass, and precocial birds tend to grow slower than altricial birds (Ricklefs, 1973). Royle et al. (1999) suggested that growth rate is associated with sibling competition. Several studies on passerines have shown that growth rates increase with nest predation rates (Cheng & Martin, 2012; Martin et al., 2011; Remeš & Martin, 2002). Sandvig, Coulson & Clegg (2019) found that birds from high latitudes tend to grow faster than birds from lower latitudes, and that birds nesting in open nests grow faster than birds nesting in enclosed nests. An inverse relationship between growth rate and locomotor performance at the interspecific level has also been proposed for songbirds (Martin, 2015). However, contradictory evidence exists at the intraspecific level. For example, Coslovsky & Richner (2011) found that within great tits (Parus major), the nestlings that grow faster tend to have longer wings at maturity. Moreover, statistical tests on the relationship between growth rate and locomotor performance in birds at the interspecific level are still lacking.

Here we empirically assess how growth rate is associated with three wing parameters of birds: flight muscle ratio, wing aspect ratio, and wing loading. The mass of flight muscles is important in determining the power that flight muscles can produce (Pennycuick, 2008). Wing loading and aspect ratio are key parameters for the aerodynamics of flight, which can be calculated based on body mass, wing span, and wing area (Pennycuick, 2008).

Materials and Methods

Data collection

Two datasets were compiled: one (81 species) to assess the relationship between growth rate and flight muscle ratio (Table S1), and the other (125 species) to assess the relationship between growth rate and wing aspect ratio and wing loading (Table S2). Body mass, development mode (precocial or altricial), nest type (open or enclosed), clutch size, latitude, and migratory status (migratory or not) were included as control variables. Data on growth rate (“K” in the logistic function) were taken from AnAge Database (Tacutu et al., 2013) and Tholon & Queiroz (2007); body mass and mass of flight muscles (m. pectoralis and m. supracoracoideus) from Wright, Steadman & Witt (2016) and Viscor & Fuster (1987); body mass, wing span, and wing area from Pennycuick (2008) and Serrano et al. (2016); development mode from Starck & Ricklefs (1998); clutch size from Myhrvold et al. (2015); nest type from Sibly et al. (2012) and Harrison & Greensmith (1993); latitude from BirdLife International & Handbook of the Birds of the World (2018); migratory status from BirdLife International (2019). Data on body mass were taken from different sources for different sets of species. Flight muscle ratio was calculated as mass of flight muscles divided by body mass; aspect ratio as wing span squared divided by wing area; wing loading as body mass divided by wing area.

Data analysis

Nine models were tested. Three models were tested to assess the relationship between growth rate and three wing parameters (flight muscle ratio, wing aspect ratio, and wing loading), respectively, while controlling for body mass, development mode, clutch size, nest type, latitude, and migratory status. Another three models were tested to assess the relationship between latitude and these three wing parameters, respectively, while controlling for body mass, development mode, clutch size, nest type, and migratory status. The last three models were tested to assess the relationship between growth rate and these three wing parameters, respectively, while controlling for body mass, development mode, clutch size, nest type, and migratory status, with latitude dropped.

All analyses were carried out in R (R Development Core Team, 2019) using the packages “ape” (Paradis, Claude & Strimmer, 2004), “caper” (Orme et al., 2018), and “phytools” (Revell, 2012). To account for phylogeny, we used 1,000 time-calibrated phylogenetic trees from birdtree.org (Jetz et al., 2012) for each dataset in our study, from which a majority rule consensus tree was derived using the function “consensus.edges” in the package “phytools”. The phylogenetic generalized least squares (PGLS) analyses with Pagel’s λ were performed using the function “pgls” in the package “caper”. Before analyses, growth rate, body mass, flight muscle ratio, wing aspect ratio, wing loading, and clutch size were log10-transformed, while the absolute values of latitude were used. The effect size “r” (for continuous variables) or “Hedges’ d” (for categorical variables) was calculated from t-values obtained from PGLS models (Nakagawa & Cuthill, 2007).

To visualize the relationship between growth rate and wing parameters, residuals of these variables were obtained by regressing them against the control variables.

Results

In order to assess the relationship between growth rate and wing parameters, we tested nine models (Tables 1–3). There are phylogenetic signals in all these models, indicating that the phylogenetic non-independence should be accounted for in these analyses.

Table 1 PGLS models of growth rate in relation to flight muscle ratio (Model 1), wing aspect ratio (Model 2), and wing loading (Model 3), while controlling for body mass, development mode, clutch size, nest type, latitude, and migratory status.

Effect size “r” was calculated for continuous variables, while “Hedges’ d” for categorical variables.

	β	P	Effect size
(r or d)	95% CI of effect size	
Model 1 (R2adjsted = 0.53, Pagel’s λ = 0.786)	
Intercept	−0.58	<0.001	–	–	
Body mass	−0.24	<0.001	−0.66	[−0.77, −0.52]	
Development mode: precocial	−0.40	<0.001	−1.52	[−2.06, −0.99]	
Clutch size	−0.02	0.818	−0.03	[−0.24, 0.19]	
Nest type: open	−4.37 × 10−4	0.991	−3.31 × 10−3	[−0.56, 0.56]	
Latitude	2.03 × 10−3	0.032	0.25	[0.03, 0.44]	
Migratory status: migratory	0.02	0.530	0.17	[−0.34, 0.69]	
Flight muscle ratio	0.41	0.007	0.31	[0.10, 0.49]	
Model 2 (R2adjsted = 0.43, Pagel’s λ = 0.643)	
Intercept	−0.94	<0.001	–	–	
Body mass	−0.19	<0.001	−0.53	[−0.65, −0.39]	
Development mode: precocial	−0.29	<0.001	−0.91	[−1.29, −0.53]	
Clutch size	0.11	0.147	0.13	[−0.04, 0.30]	
Nest type: open	0.05	0.116	0.33	[−0.07, 0.73]	
Latitude	2.86 × 10−3	0.001	0.29	[0.12, 0.44]	
Migratory status: migratory	0.06	0.115	0.37	[−0.08, 0.82]	
Aspect ratio	−0.11	0.593	−0.05	[−0.22, 0.13]	
Model 3 (R2adjsted = 0.44, Pagel’s λ = 0.667)	
Intercept	−1.16	<0.001	–	–	
Body mass	−0.25	<0.001	−0.47	[−0.59, −0.32]	
Development mode: precocial	−0.32	<0.001	−1.00	[−1.38, −0.61]	
Clutch size	0.13	0.073	0.16	[−0.01, 0.33]	
Nest type: open	0.05	0.122	0.32	[−0.07, 0.72]	
Latitude	2.67 × 10−3	0.002	0.28	[0.11, 0.43]	
Migratory status: migratory	0.04	0.202	0.30	[−0.15, 0.75]	
Wing loading	0.18	0.090	0.16	[−0.02, 0.32]	

Table 2 PGLS models of latitude in relation to flight muscle ratio (Model 4), wing aspect ratio (Model 5), and wing loading (Model 6), while controlling for body mass, development mode, clutch size, nest type, and migratory status.

	β	P	Effect size
(r or d)	95% CI of effect size	
Model 4 (R2adjsted = 0.13, Pagel’s λ = 0.418)	
Intercept	32.61	0.055	–	–	
Body mass	7.82	0.027	0.25	[0.04, 0.45]	
Development mode: precocial	7.10	0.257	0.29	[−0.19, 0.77]	
Clutch size	8.17	0.353	0.11	[−0.11, 0.32]	
Nest type: open	5.97	0.212	0.37	[−0.19, 0.94]	
Migratory status: migratory	8.39	0.045	0.55	[0.03, 1.08]	
Flight muscle ratio	17.32	0.315	0.12	[−0.10, 0.33]	
Model 5 (R2adjsted = 0.16, Pagel’s λ = 0.562)	
Intercept	62.39	0.002	–	–	
Body mass	8.03	0.005	0.26	[0.08, 0.41]	
Development mode: precocial	14.64	0.015	0.47	[0.10, 0.84]	
Clutch size	13.18	0.086	0.16	[−0.02, 0.32]	
Nest type: open	1.09	0.748	0.07	[−0.33, 0.46]	
Migratory status: migratory	11.88	0.001	0.76	[0.31, 1.22]	
Aspect ratio	−54.19	0.010	−0.23	[−0.39, −0.06]	
Model 6 (R2adjsted = 0.14, Pagel’s λ = 0.56)	
Intercept	1.10	0.920	–	–	
Body mass	−0.09	0.984	−1.90 × 10−3	[−0.18, 0.17]	
Development mode: precocial	7.79	0.198	0.25	[−0.12, 0.61]	
Clutch size	19.47	0.009	0.24	[0.07, 0.40]	
Nest type: open	0.34	0.919	0.02	[−0.38, 0.42]	
Migratory status: migratory	8.25	0.022	0.54	[0.09, 0.99]	
Wing loading	22.73	0.041	0.19	[0.01, 0.35]	

Table 3 PGLS models of latitude in relation to flight muscle ratio (Model 7), wing aspect ratio (Model 8), and wing loading (Model 9), while controlling for body mass, development mode, clutch size, nest type, and migratory status, with latitude dropped.

	β	P	Effect size
(r or d)	95% CI of effect size	
Model 7 (R2adjsted = 0.51, Pagel’s λ = 0.792)	
Intercept	−0.52	0.001	–	–	
Body mass	−0.22	<0.001	−0.63	[−0.75, −0.48]	
Development mode: precocial	−0.38	<0.001	−1.42	[−1.95, −0.89]	
Clutch size	4.84 × 10−4	0.995	7.36 × 10−4	[−0.22, 0.22]	
Nest type: open	0.01	0.833	0.06	[−0.50, 0.62]	
Migratory status: migratory	0.04	0.206	0.35	[−0.17, 0.86]	
Flight muscle ratio	0.43	0.006	0.31	[0.10, 0.50]	
Model 8 (R2adjsted = 0.37, Pagel’s λ = 0.717)	
Intercept	−0.79	<0.001	–	–	
Body mass	−0.16	<0.001	−0.46	[−0.59, −0.31]	
Development mode: precocial	−0.25	<0.001	−0.71	[−1.09, −0.34]	
Clutch size	0.17	0.035	0.19	[0.02, 0.36]	
Nest type: open	0.05	0.114	0.33	[−0.07, 0.73]	
Migratory status: migratory	0.09	0.017	0.56	[0.11, 1.01]	
Aspect ratio	−0.25	0.237	−0.11	[−0.28, 0.07]	
Model 9 (R2adjsted = 0.38, Pagel’s λ = 0.733)	
Intercept	−1.17	<0.001	–	–	
Body mass	−0.24	<0.001	−0.44	[−0.57, −0.29]	
Development mode: precocial	−0.30	<0.001	−0.84	[−1.22, −0.46]	
Clutch size	0.20	0.009	0.24	[0.06, 0.40]	
Nest type: open	0.05	0.127	0.32	[−0.08, 0.72]	
Migratory status: migratory	0.06	0.075	0.42	[−0.03, 0.87]	
Wing loading	0.24	0.028	0.20	[0.03, 0.36]	

When controlling for body mass, development mode, clutch size, nest type, latitude, and migratory status, among the three wing parameters, only flight muscle ratio was significantly correlated with growth rate (P = 0.007, effect size = 0.31), while wing aspect ratio (P = 0.593, effect size = −0.05) and wing loading (P = 0.090, effect size = 0.16) were not (Models 1–3 in Table 1). Birds with higher flight muscle ratios tended to grow faster than birds with smaller flight muscle ratios (Fig. 1). In all these three models, latitude was significantly correlated with growth rate (P = 0.032, effect size = 0.25 in model 1; P = 0.001, effect size = 0.29 in model 2; P = 0.002, effect size = 0.28 in model 3); birds from higher latitudes tended to grow faster than birds from lower latitudes.

Figure 1 Bivariate plots showing the relationships between growth rate and wing parameters in birds.

(A) residual log10 (growth rate) vs. residual log10 (flight muscle ratio); (B) residual log10 (growth rate) vs. residual log10 (aspect ratio); (C) residual log10 (growth rate) vs. residual log10 (wing loading). The lines are simple regressions between the residuals. In (A) the control variables are body mass, development mode, clutch size, nest type, latitude, and migratory status, while in (B) and (C) the control variables are body mass, development mode, clutch size, nest type, and migratory status, with latitude dropped.

Latitude was significantly correlated with wing aspect ratio (P = 0.010, effect size = −0.23) and wing loading (P = 0.041, effect size = 0.19), but not with flight muscle ratio (P = 0.315, effect size = 0.12), when controlling for body mass, development mode, clutch size, nest type, and migratory status (Models 4–6 in Table 2).

After dropping latitude from the models, growth rate was significantly correlated with flight muscle ratio (P = 0.006, effect size = 0.31) and wing loading (P = 0.028, effect size = 0.20, Fig. 1), but not with wing aspect ratio (P = 0.237, effect size = −0.11) (Models 7–9 in Table 3). Fast-growing birds tended to have higher flight muscle ratios and higher wing loadings than slow-growing birds.

Discussion

Our results show that fast-growing birds tended to have higher flight muscle ratios and higher wing loadings than slow-growing birds, which suggests that the relationship between growth rate and flight ability in birds is more complex than a simple tradeoff. It has been shown that the takeoff ability of birds is largely dependent on flight muscle ratio (Hartman, 1961; Marden, 1987). Birds with higher flight muscle ratios can provide larger mass-specific lift force and take off more steeply than birds with lower flight muscle ratios. The positive correlation between growth rate and flight muscle ratio suggests that fast-growing birds tend to have better escape performance from predators than slow-growing birds. In the fixed-wing model of maneuvering performance, the radius of turn is proportional to wing loading; that is, birds with lower wing loadings can make turns of smaller radii (Norberg & Norberg, 1971; Norberg, 1990; Pennycuick, 2008). However, it has been suggested that birds with high wing loadings can also effect turns of small radii, but requiring slowing and flapping, which is energetically more expensive (Warrick, Dial & Biewener, 1998; Warrick, Bundle & Dial, 2002). In other words, wing loading is associated with efficiency of maneuvering flight (Warrick, Bundle & Dial, 2002). The positive correlation between growth rate and wing loading suggests fast growth of birds may negatively affect the efficiency of maneuvering flight. Aspect ratio reflects the efficiency of flight; an increase of aspect ratio can increase the lift and reduce the drag (Norberg, 1990; Pennycuick, 2008). The lack of significant correlation between growth rate and aspect ratio suggests that growth rate and efficiency of flight are likely to be disconnected.

While the tradeoff caused by limited resources can explain growth rate’s negative association with wing area (birds with smaller wing areas have higher wing loadings), the mechanisms underlying the positive correlation between growth rate and the size of flight muscles remain to be explored. Wright, Steadman & Witt (2016) showed that island birds tend to evolve smaller flight muscles and found a positive correlation between the size of flight muscles and predation pressure, when using raptorial species richness and the presence of mammalian predators as proxies for predation pressure. Sandvig, Coulson & Clegg (2019) showed that among altricial birds, island birds tend to grow slower than continental birds, though the relationship is marginally non-significant. A positive correlation between growth rate and nest predation rate has been demonstrated in previous studies of passerines (Cheng & Martin, 2012; Martin et al., 2011; Remeš & Martin, 2002). These studies suggest that predation pressure can be a potential factor that drives the correlated evolution of growth rate and the size of flight muscles in birds.

The positive relationship found between growth rate and latitude is consistent with previous studies (Martin, 2015; McCarty, 2001; Ricklefs, 1968, 1976; Sandvig, Coulson & Clegg, 2019). Moreover, latitude may confound growth rate’s association with wing loading. Martin (2015) suggested that the slower growth of tropical birds is associated with enhanced flight performance after fledging than temperate birds. The positive association of latitude with wing loading and the negative association of latitude with wing aspect ratio suggest that tropical birds may be more efficient in maneuvering flight and flight in general.

In extant birds, the sternal keel serves as the attachment of flight muscles (i.e., m. supracoracoideus and m. pectoralis) and the sternal keel length is positively correlated with the mass of flight muscles (Wright, Steadman & Witt, 2016). By contrast, in the earliest fossil birds, for example, Archaeopteryx, an ossified sternal keel is absent (Zheng et al., 2014), and an enlargement of the sternal keel along the lineage leading to crown birds has been well documented (O’Connor et al., 2015b; Zheng et al., 2014, 2012). However, how the absence or the small size of the sternal keel in early birds could affect the size of flight muscles remains to be elucidated (Mayr, 2017; O’Connor et al., 2015a; Olson & Feduccia, 1979). Recent prolific studies suggest that growth rates of extinct taxa can be estimated from their bone histology (Cubo et al., 2012; Erickson, 2005, 2014; Erickson et al., 2009; Erickson, Rogers & Yerby, 2001; Padian, De Ricqlès & Horner, 2001). Accordingly, our finding of the positive correlation between growth rate and the size of flight muscles suggests that bone microstructures may also be associated with the size of flight muscles. Further investigations on the relationship between bone histology and the size of flight muscles, and possibly other flight-related parameters, may provide a new avenue to understanding the early evolution of flight and change in growth rate.

Conclusions

Our study shows that growth rate and flight ability are correlated in avian evolution, and their relationship is more complex than a simple tradeoff as proposed in previous studies. Fast-growing birds tended to have higher flight muscle ratios and higher wing loadings, which means that fast-growing birds may have better takeoff performance, but lower efficiency in maneuvering flight. Besides wings, legs contribute greatly to the locomotion of birds and are important for birds to occupy different habitats (Habib & Ruff, 2008; Stoessel, Kilbourne & Fischer, 2013; Zeffer, Johansson & Marmebro, 2003). Moreover, wings and legs are highly linked during avian evolution (Allen et al., 2013; Heers & Dial, 2015; Zhao, Liu & Li, 2017). Further studies on the relationship between growth rate and hindlimb performance will provide more insights into the evolution of birds.

Supplemental Information

Supplemental Information 1 Raw data.

Click here for additional data file.

Supplemental Information 2 R code and phylogenetic trees.

Click here for additional data file.

We are grateful to Prof. Zhonghe Zhou for discussion and Dr. Zhenchao Wang for help with ArcGIS.

Additional Information and Declarations

Competing Interests

Author Contributions

Data Availability

The authors declare that they have no competing interests.

Tao Zhao conceived and designed the experiments, performed the experiments, analyzed the data, prepared figures and/or tables, authored or reviewed drafts of the paper, and approved the final draft.

Zhiheng Li conceived and designed the experiments, authored or reviewed drafts of the paper, and approved the final draft.

The following information was supplied regarding data availability:

The raw data, phylogenetic trees, and R code are available as Supplemental Files.

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
