# Peer review of "Growth rate and locomotor performance tradeoff is not universal in birds"

_PeerJ, doi:10.7717/peerj.8423_

## Round 0.1 · original submission · Major Revisions

We have received two reviews on your manuscript. Both reviewers found positive aspects about the study, but reviewer 1 also raised a number of points that deserve further revisions.

In particular, the statistical modelling approach needs further clarifications and justification. Collinearity among variables may be an issue here and it needs to be assessed. I was also surprised than nonlinear relationships were not considered (ex. quadratic relationships between mass and growth, etc.) – this should be addressed in the next version. Also, as suggested by reviewer 1, additional relevant variables such as habitat type and migration distance may need to be considered. The approach for modelling predation pressure and flight style of birds may also need to be re-evaluated.

The results section of the manuscript needs to be greatly improved by including effect sizes/correlations and r2 and not just p-values. The discussion also needs to be revised and conclusions toned down in some instances.

Finally, the revisions should also integrate all other comments and text edits made by reviewers. The supplementary material formatting (especially Table S3) also needs to be upgraded.

Reviewer 1 ·

Basic reporting

No comment.

Experimental design

No comment.

Validity of the findings

Some of the models may have issues with collinearity, as body mass, linear wing dimensions, and flight muscle mass are highly related. For example, body mass and wing area will be highly collinear, as larger species generally have bigger absolute wing spans than smaller species. This type of analysis may lead to multicollinearity, and efforts to account for multicollinearity should be used. One way around this may be to combine these variables. For example, wing loading (mass per unit wing area) would be a natural choice for analysis of wing area and growth rates, as wing loading is important in aerodynamic theory. You may also want to see Vásági et al. (2016; Evol Biol 43:48-59) for other ways to take into account scaling of morphological traits with body size.

Some of the conclusions drawn by the manuscript are a bit strong, and may be more speculative. The language around these Conclusion section (line 200-208) should reflect this as such. For example, the authors conclude that fast growing birds, because of their smaller wings, are not as maneuverable as species with larger wings. However, the maneuverability is inferred based on morphological data, rather than quantifying maneuverability in some fashion. Thus, the final and main conclusion (Lines 200-208) is at best speculation. I would suggest that these conclusions be tempered, as there is very little comparative information regarding maneuverability in birds, except for hummingbirds (Dakin et al. 2018, Science).

Additional comments

This paper compares growth rates and morphological variation in bird species, and whether there are compromises in the relative size of the wings and the supporting musculature with increasing growth rates. The authors assembled a comprehensive database from the literature, including wing morphology, latitude, muscle mass, growth rates, and nesting characteristics. Their models suggest that faster growing birds tend to have smaller wings and bigger muscles than slower growing species. The effort of assembling this database is large and I commend the authors on getting the data on such a large number of species together. The research question is well defined, and that data collected would provide some insights into the question. However, I do have some concerns regarding the statistical models that were initially chosen, and whether other variables should be taken into account. I have outlined my concerns below:

One aspect of this study that needs to be closely considered is the ecology of the birds, and whether different habitat types (eg. forest versus savannah versus aquatic) or migratory tendency (nonmigratory, short-distance, long-distance migrant) also played an important role in wing morphological variation. For example, Norberg and Rayner 1987 (Phil. Trans. R. Soc. B. vol. 316 issue 1179) have noted differences in wing morphology in bats, with bats in “cluttered” environments tending to have shorter wings compared to species that tend to fly in open environments. This suggests that flight environment plays an important role in determining relative wing proportions, but also suggests that shorter wings may provide more maneuverability compared to longer wings. Migration is another consideration for morphological variation, as it is predicted that migrants have wings that permit fast and efficient flight (Lindhe Norberg 2002, J Morph 252(1): 52-81). Migrants also tend to breed at higher latitudes, but not all high latitude birds are migrants. Thus, migratory patterns within species should be considered Vásági et al. (2016; Evol Biol 43:48-59) may be a good resource that links morphology to migratory distance. I recommend including habitat type and migration distance (either as a continuous variable, or as discrete “non-migratory versus migratory”) in the models.

The way predation pressure was modeled is a bit peculiar as it is subsumed in the "nest type" category. For example, seabirds such as albatrosses, tend to nest on relatively isolated islands with lower predation pressure compared to mainland species, but are classified as cup type which is considered high predation risk. Thus, while the cup versus enclosed analysis may apply to some bird species that nest on the mainland where predation pressure is high, it may be more difficult to apply to species with relatively lower predation pressure, such as very large bird species or birds that nest on isolated islands. I would recommend that predation pressure be quantified in some other way, besides nest type.

Different flight styles (eg. flapping versus soaring) are also not incorporated into the analysis. Soaring and gliding flight styles are associated with a range of morphological adaptations such as long and thin wings that maximize glide ratio for extended distances. Thus, inclusion of birds adapted for soaring with birds relying primarily on flapping flight may confound some of the analyses. I would recommend splitting the analysis into flapping flight and soaring flight, for example, or only including birds that use flapping flight.

This also brings into question whether birds that rely on soaring flight have relatively smaller muscles than birds that rely on flapping flight, and the scaling relationship between muscle mass and body size in soaring versus flapping birds. This is a "bonus" question that should be addressed in your manuscript, and I believe you have enough data to do so.

Reviewer 2 ·

Basic reporting

no comment

Experimental design

no comment

Validity of the findings

no comment

Additional comments

This is an interesting study. I have only some few minor comments. I have also provided some minor corrections on the ms itself.
I have provided an alternative title.

Intra and interspecific results are mixed sometimes without clearly differentiating between the two (e.g. L 69-71).
L90: In L78 it was flight muscle mass, not ratio. Please clarify which was the variable tested.
L97: Body mass seems to be taken from different sources. Please clarify (I suppose it is because it is for different sets of species).
L136-144: I have not found information on the amount of variance explained by the models. P-values are not enough.

Annotated reviews are not available for download in order to protect the identity of reviewers who chose to remain anonymous.

---

## Round 0.2 · accepted · Accept

The reviewer and I are both satisfied with the modifications made on the manuscript.

Reviewer 1 ·

Basic reporting

No comment

Experimental design

No comment

Validity of the findings

No comment

Additional comments

The authors have provided an updated manuscript, and I am pleased with the changes that have been made. The inclusion of effect sizes, p-values, and model summaries in the main body of the text has greatly improved readability and the ease of interpretation of the models. Furthermore, the use of more mass-independent metrics of wing morphology and muscle size have improved the statistics. Overall, I am pleased with the changes that the authors have provided.